



# Comparative experimental validation of microwave hyperspectral atmospheric soundings in clear-sky conditions

Lei Liu[1], Natalia Bliankinshtein[2], Yi Huang[1], John R. Gyakum[1], Philip M. Gabriel[3], Shiqi Xu[2], Mengistu Wolde[2]

[1]Department of Atmospheric and Oceanic Sciences, McGill University, Montreal, Quebec, Canada
[2]Flight Research Laboratory, National Research Council Canada, Ottawa, Ontario, Canada
[3]Horizon Science and Technology, Wolfville, Nova Scotia, Canada

*Correspondence to*: Lei Liu (lei.liu5@mail.mcgill.ca)

**Abstract.** Accurate observations of atmospheric temperature and water vapor profiles are essential for weather forecasting
and climate change detection. Hyperspectral radiance measurements afford a useful means to retrieve these thermodynamic variable fields, by harnessing the rich information contained in the electromagnetic wave spectrum of the atmospheric radiation. Compared to infrared radiometry, microwave radiometry holds the ability to penetrate clouds and potentially achieve an all-sky thermodynamic retrieval. Recent technological advancements have enabled the development of a hyperspectral microwave radiometer, the High Spectral Resolution Airborne Microwave Sounder (HiSRAMS), which we
employ in this study to retrieve the atmospheric temperature and water vapor profiles under the clear-sky condition, in comparison with an infrared hyperspectrometer, the Atmospheric Emitted Radiance Interferometer (AERI). HiSRAMS and AERI measurements under different viewing geometries have been acquired and compared for atmospheric retrieval. When both instruments are placed on the ground to acquire zenith-pointing measurements, the infrared hyperspectral measurements exhibit higher information content and greater vertical resolution for temperature and water vapor retrievals than the
microwave hyperspectral measurements. A synergistic fusion of HiSRAMS and AERI measurements from the air and ground, respectively, is tested. This "sandwich" sounding of the atmosphere takes advantage of the complementary information contents of the two instruments and is found to notably improve retrieval accuracy.



## 1 Introduction

Temperature and water vapor concentration vertical profiles are fundamental thermodynamic variable fields and play a crucial role in diverse meteorological applications, ranging from extreme weather forecasting to long-term climate change detection (Guo et al., 2020; Langland and Baker, 2004; Laroche and Sarrazin, 2010; Thorne et al., 2011; Wang et al., 2016). Multiple methods are employed to measure these profiles, including direct, in situ measurements through radiosondes and aircrafts (Bliankinshtein et al., 2023; Durre et al., 2006; Petzold et al., 2015; Zhou et al., 2021), indirect, remote sensing

measurements obtained from spectrally resolved radiance (Aires et al., 2015; Loveless, 2021; Turner and Blumberg, 2018; Susskind et al., 2003, 2010; Pougatchev et al., 2009), and data assimilation (Gelaro et al., 2017; Hersbach et al., 2020).

Direct measurements offer precise temperature and water vapor concentration profiles but have limited spatial and temporal coverage, in contrast to indirect, remote sensing (spectral) measurements, essential for regional and global weather and climate analyses. In these measurements, the temperature and water vapor information is typically encoded in the

atmospheric radiance spectra. An algorithm is required to retrieve this information from atmospheric emission, absorption, or scattering features of the atmosphere across various frequency ranges. Hyperspectral measurements are particularly useful in this application because the high spectral resolution translates to richer information content. Several hyperspectral measurement methods were undertaken for atmospheric temperature and water vapor soundings; these can be categorized in terms of deployment platform (e.g., ground-based, airborne, or spaceborne) or frequency range (infrared, microwave, or

other radiometers).

Clouds cover more than half of the Earth's surface (Stubenrauch et al., 2010), wielding a significant impact on hyperspectral temperature and water vapor retrievals, primarily due to their masking effect (Mcnally and Watts, 2003). In the infrared spectral range, clouds tend to be optically thick, effectively obscuring the atmosphere and preventing the retrieval of the target (atmospheric temperature and water vapor) behind the cloud layer. In contrast, microwave signals penetrate clouds,

enabling the retrieval of temperature and water vapor profiles of the entire atmospheric column and a true all-sky sounding of the atmosphere.

For this reason, microwave radiometers have been a subject of active studies for decades(Aires et al., 2015; Blackwell et al., 2010; Hilliard et al., 2013; Smith et al., 2021). In the past, microwave radiometers typically had several or, in rare cases, dozens of channels, limiting the vertical resolution of temperature and water vapor retrievals. However, recent advancements

in microwave Fast Fourier Transfer (FFT) filter techniques have led to the development of very high spectral resolution microwave radiometers, offering similar numbers (thousands) of channels to infrared hyperspectral radiometers and the potential for an information content boost. In this study, we deploy an airborne hyperspectral microwave spectrometer, the High Spectral Resolution Airborne Microwave Sounder (HiSRAMS), developed by an international team (Auriacombe et al., 2022; Bliankinshtein et al., 2023). HiSRAMS is equipped with two radiometers and operates in the microwave spectral

ranges covering two absorption bands of oxygen and water vapor respectively. It can be configured to measure single-



polarized or dual-polarized radiance. As an airborne instrument, it can provide both zenith-pointing (looking up) and nadir-pointing (looking down) measurements (Bliankinshtein et al., 2023).

A ground-based infrared radiometer, the Atmospheric Emitted Radiance Interferometer (AERI), was also utilized in this study to compare with HiSRAMS in terms of temperature and water vapor concentration retrieval performance. AERI is a
well-tested infrared interferometer, which measures downwelling radiance emitted from the atmosphere between 520 and 3200 cm$^{-1}$ with a spectral resolution of 0.5 cm$^{-1}$ (Knuteson et al., 2004a, 2004b). AERI has been used to retrieve temperature and water vapor vertical profiles using the Optimal Estimation method with acceptable accuracy, particularly for near-surface profiles (Feltz et al., 2003; Feltz et al., 1998; Turner and Löhnert, 2014; Turner and Blumberg, 2018; Turner et al., 2000).

While the primary advantage of microwave radiometers lies in their ability to retrieve in cloudy-sky conditions, this study focuses initially on clear-sky retrievals. The study has two main objectives: 1) to test the abilities of HiSRAMS and AERI to retrieve temperature and water vapor profiles while they are both placed on the ground taking zenith measurements, which provides identical conditions for assessing the two hyperspectral instruments in terms of their retrieval performance; 2) to test a synergistic retrieval combining AERI's ground-based zenith measurements with HiSRAMS' airborne nadir
measurements, which allows the exploration of the complementary information from the two instruments operating in different spectral ranges and observing from different viewing geometries.

## 2 Field campaigns

Two field campaigns were conducted to retrieve vertical profiles of temperature and water vapor concentration profiles using AERI and HiSRAMS. The first campaign, denoted as FC2022, took place on December 9, 2022; the second campaign,
FC2023, on February 11, 2023. During both campaigns, AERI and HiSRAMS measurements were collected alongside radiosonde data. FC2022 was a ground-based campaign, with AERI and HiSRAMS solely acquiring zenith-pointing measurements from the ground. In contrast, FC2023 included ground measurements of HiSRAMS and AERI and flight measurements of HiSRAMS at various observational altitudes. A thorough description of the field campaigns, particularly the radiative closure analysis of AERI and HiSRAMS measurements, can be found in Liu et al. (2023). This study focuses
on the FC2023 campaign, during which the highest observational altitude of HiSRAMS reached 6.8 km, which allows us to validate the temperature and water vapor concentration retrievals within the troposphere against the measurements of radiosonde and aircraft. The HiSRAMS measurements used in this study are single-polarized measurements.

## 3 Retrieval algorithms

Our retrieval algorithms, employing HiSRAMS and AERI data, are based on the optimal estimation method (Rodgers, 2000;
Loveless, 2021; Turner and Blumberg, 2018). These instruments measure the radiance, $\boldsymbol{y}$, received by their respective





detectors. A forward model, $F$, is utilized to simulate the radiance $F(x)$ under the atmospheric state conditions, $x$, shown in Eq. 1. The AERI forward model we adopt is the Line-by-Line Radiative Transfer Model Version 12.9 (LBLRTM) and the HiSRAMS forward model has been developed and validated by Bliankinshtein et al. (2019). More details of the forward models can be found in Liu et al. (2023). In this study, the state vector $x$ encapsulates vertical profiles of temperature and

water vapor concentration. $\varepsilon$ represents the error including both the measurement error and the forward model error.

$$y = F(x) + \varepsilon \tag{1}$$

Linearizing the relationship between $x$ and $y$ at a reference state $x_0$:

$$y = F(x_0) + \frac{\partial F(x)}{\partial x}(x - x_0) + \varepsilon = F(x_0) + K(x - x_0) + \varepsilon \tag{2}$$

Here $K = \frac{\partial F(x)}{\partial x}$ is the Jacobian matrix, representing the sensitivity of the forward model to the state vector. $K_{AERI}$ and

$K_{HiSRAMS}$ are obtained by the analytical Jacobian method using their respective forward models. This linearization is a good approximation for the temperature values under question. However, for water vapor, we find that linearization works better for the logarithm of the water vapor concentration (Huang and Bani Shahabadi, 2014). Thus, $x_T = T$ and $x_q = log(q)$, where $T$ represents atmospheric temperature in units of K and $q$ represents atmospheric water vapor in units of ppmv.

The objective of the retrieval is to infer $x$ from $y$. To address the nonunique correspondence between the atmospheric state

conditions and the radiance, the Optimal Estimation method minimizes the cost function $J$ or optimizes a posteriori. Applying the Levenberg-Marquardt iteration method and using a multiplier $\gamma$ to stabilize the iteration processes by assigning varying weights between the measurement and the a priori (Rodgers, 2000), the state vector at iteration step $j + 1$, $x_{j+1}$, is determined by Eq. 3. The maximum step number is arbitrarily set to be 20. For all the retrieval cases in this study, the state vector converges before the maximum iteration step.


$$x_{j+1} = x_j + \left[ K_j{}^T S_e{}^{-1} K_j + (1 + \gamma) S_a{}^{-1} \right]^{-1} \left\{ K_j{}^T S_e{}^{-1} [y - F(x_j)] - S_a{}^{-1}(x_j - x_a) \right\} \tag{3}$$

Here, $S_e$ and $S_a$ are the measurement and a priori error covariance matrix, representing the covariance of the measurement error at different observational channels and of the state vectors at different vertical levels, respectively. The diagonal elements of $S_e$ and $S_a$ are the variance of the errors and the off-diagonal elements are the inter-channel or inter-layer covariance of the errors. Considering that the covariance of the AERI measurement errors are relatively small (Turner and

Blumberg, 2018), the off-diagonal elements of $S_{e,AERI}$ are set to be 0. The square root of the diagonal components of $S_{e,AERI}$ and $S_{e,HiSRAMS}$ are shown in Figure S1. AERI radiance has a relatively smaller measurement uncertainty in the window band (800 – 1200 cm$^{-1}$), although it translates to large brightness temperature uncertainty because the radiance signal in this band is typically small, especially in the clear-sky condition.

The a priori dataset consists of the hourly-mean temperature and water vapor concentration profiles from the fifth generation

European Centre for Medium-Range Weather Forecasts atmospheric reanalysis dataset, ERA5 (Hersbach et al., 2020). The hourly-mean profiles in Februaries from 1944 to 2022 at 9 grid boxes centered around the Ottawa International Airport (latitude: 45.32°, longitude: -75.66°) were collected to capture the temporal and spatial variability of the atmospheric state



variables. The vertical coordinate we adopted in this study is altitude. Thus, both ERA5 hourly-mean surface level and pressure level (37 levels) data are utilized to form the a priori dataset, which has 38 vertical levels in total (shown in Figure S2). The thickness of the layers is 1.26 km on average, with the lowermost few layers being centered at 0.13, 0.24, 0.44, 0.64, 0.85, 1.07, 1.29, and 1.51 km, respectively. The reason that we did not use higher vertical resolution for the a priori is that the vertical resolution of the retrieval is already limited by the Jacobian matrix and the measurement error covariance matrix (details in the following analysis). The covariance matrix $S_a$ used in this study is shown in Figure S3 and the correlation matrix $C_a$, which represents the correlation coefficient in the a priori dataset, is shown in Figure S4. The first guess of the state vector, $x_1$, is the mean profile of all the hourly-mean profiles in the a priori dataset.

Ideally, the Jacobian matrix should be updated at every iteration step. However, the calculation of the Jacobian matrix is the most computationally expensive step in the retrieval. Owning to the relatively smaller change of the atmospheric state vectors after iteration step 2, which results in a relatively smaller change of the Jacobian matrix, we set the AERI Jacobian matrix for all the following iteration steps to that from step 2, $K_{AERI,2}$. The calculation of the $K_{HiSRAMS}$ is fast so it is updated in every iteration step. The AERI analytical Jacobian in high spectral resolution was obtained first and then it was convolved with the AERI scan function to match AERI channels. The AERI and HiSRAMS Jacobians and the standard deviation of the state variables at selected levels are illustrated in Figures S5, S6, and S7.

In Eq. 3, we use a regularization parameter, $\gamma$, to weigh the measurement and a priori according to their error magnitudes. It is set to be a large value at the first step, decreasing with iterations until the convergence criterion (described below) is met. A set of sensitivity tests was performed to find the appropriate initial value of $\gamma$ and how it should change with iterations. The final setting of the initial value of $\gamma$ is 10000. For each iteration, if the cost function $J$ shown in Eq. 4 is increasing, $\gamma$ is increased by 10 times and the state vector is updated based on the new $\gamma$ until $J$ is smaller than that for the previous step. While $J$ is decreasing and $\gamma$ is larger than 1, $\gamma$ is decreased by 10 times for the next iteration step.

$$J = \left(x_j - x_a\right)^T S_a^{-1} \left(x_j - x_a\right) + \left[y - F(x_j)\right]^T S_e^{-1} \left[y - F(x_j)\right] \tag{4}$$

The information content of the retrievals finds common usage in assessing the retrievability of the atmospheric state variables. The averaging kernel matrix, $A$, defined as the derivative of the estimated state vectors to the true state vectors, can be derived based on $K$, $S_e$ and $S_a$, shown in Eq. 5. The Degrees of Freedom for Signal (DFS), which is the trace of the averaging kernel matrix, are adopted to quantify the information content of the retrievals (Eq. 6). A higher value of DFS means that more information content can be retrieved. The DFS for each retrieved vertical level tells us how many pieces of independent information we can get for this specific level. Ideally, for each vertical level, the DFS equals to 1. Yet due to various limiting factors, including the measurement errors and the covariance between different levels, the DFS is normally below 1.

$$A = \left(K_j^T S_e^{-1} K_j + S_a^{-1}\right)^{-1} K_j^T S_e^{-1} K_j \tag{5}$$

$$DFS = trace(A) \tag{6}$$



Another relevant measure of the retrieval performance is the retrieval uncertainty. The posterior error covariance matrix, $S$, is defined in Eq. 7. The square root of the diagonal elements of $S$ provides the $1\sigma$ uncertainty in the retrieved atmospheric state variables. Both $A$ and $S$ are iterated over the steps and are impacted by the value of $\gamma$. In order to have fair comparison between different retrieval cases, the final values of $A$ and $S$ are determined when $\gamma$ is 0.

$$S = \left(K_j{}^T S_e{}^{-1} K_j + S_a{}^{-1}\right)^{-1} \tag{7}$$

For all these matrices, the dimensions are only based on the dimension of the vertical level ($n_{level}$) and the dimension of the instrumental channels ($n_{AERI}$ and $n_{HiSRAMS}$). In this study, we retrieve temperature and water vapor vertical profiles simultaneously. Thus, $x$ equals to $\begin{bmatrix} x_T \\ x_q \end{bmatrix}$ with a dimension of $38 \times 2 = 76$. Because HiSRAMS is an airborne instrument, the light path of the HiSRAMS may be limited by the observational altitude when pointing directly downward (nadir-pointing). Thus, $n_{level}$ varies for different case studies (detailed in the following sections). In order to test the full potential of AERI

and HiSRAMS to retrieve temperature and water vapor concentration profiles, all the instrumental channels are kept, resulting $n_{AERI} = 2490$ and $n_{HiSRAMS} = 2850$ (including the measurements of both spectrometers of HiSRAMS).

The retrieval is considered to be converged when the convergence criteria, shown in Eq. 8 are met:

$$d_{x,j}{}^2 = \left(x_j - x_{j+1}\right)^T \left(K_j{}^T S_e{}^{-1} K_j + S_a{}^{-1}\right)\left(x_j - x_{j+1}\right) < \min\left\{d_{threshold}{}^2, \frac{n_{level}}{20}\right\} \tag{8}$$

Where $d_{x,j}{}^2$ represents the change of uncertainty in state vector space. The threshold of this parameter, $d_{threshold}{}^2$, is

determined when the temperature change between two iteration steps equals 0.5K ($\Delta x_T = 0.5K$), the water vapor concentration change between two iteration steps equals a 10% change in water vapor concentration [$\Delta x_q = \log(1.1)$]. These values represent the expected accuracy of the variables. Note that $d_{threshold}{}^2$ varies with each iteration due to updates in the Jacobian matrix.

To assess the sensitivity of the measurements while accounting for the state vector variability and the measurement

uncertainty together, we derived a metric, the Signal-to-Noise Ratio (SNR) as defined in Eq. 9, where $\sigma(x)$ represents the standard deviation of the state vector in the a priori dataset. Based on their SNR values, we can have a fair comparison between different measurements, i.e., AERI and HiSRAMS in this study.

$$SNR = \frac{K\sigma(x)}{\sqrt{S_{e,diag}}} \tag{9}$$

In this study, we obtained temperature and water vapor retrievals based on single instruments (AERI or HiSRAMS) and joint

instruments (AERI and HiSRAMS), respectively. Regardless of the retrieval cases, the dimensions of $S_a$, $S$, and $A$ are all $n_{level} \times n_{level}$. $S_a$, $S$, and $A$ all have a similar matrix structure: the upper-left sub-matrix and the lower-right sub-matrix are for temperature and water vapor respectively. As a result, we can separate the information of temperature and water vapor. When retrieving the temperature and water vapor vertical profiles using either AERI or HiSRAMS alone, the dimensions of $S_e$ and $K$ are $n_{instrument} \times n_{instrument}$ and $n_{instrument} \times n_{level}$, respectively, where 'instrument' refers to either AERI or

HiSRAMS. For joint retrieval:



$$y_{joint} = \begin{bmatrix} y_{AERI} \\ y_{HiSRAMS} \end{bmatrix} \tag{10}$$

$$S_{e,joint} = \begin{bmatrix} S_{e,AERI} & 0 \\ 0 & S_{e,HiSRAMS} \end{bmatrix} \tag{11}$$

$$K_{joint} = \begin{bmatrix} K_{AERI} \\ K_{HiSRAMS} \end{bmatrix} \tag{12}$$

The dimensions of $y_{joint}$, $S_{e,joint}$, and $K_{joint}$ are $(n_{AERI} + n_{HiSRAMS}) \times 1$, $(n_{AERI} + n_{HiSRAMS}) \times (n_{AERI} + n_{HiSRAMS})$, and $(n_{AERI} + n_{HiSRAMS}) \times n_{level}$ respectively.

## 4 Ground-based HiSRAMS and AERI retrievals

Ground measurements of AERI and HiSRAMS were obtained in campaign FC2023. Simultaneous temperature and water vapor retrievals were performed for single instruments (AERI or HiSRAMS) to compare their retrieval performance. For ground-based retrievals, the number of the vertical levels is 38. Thus, $n_{level}$ is 76.

### 4.1 Temperature retrieval

The total DFS corresponding to temperature retrievals across the entire atmospheric column are quantified to be 9.52 and 5.27 for AERI and HiSRAMS, respectively. Notably, AERI exhibits a higher information content in temperature when compared to HiSRAMS. To further elucidate the distribution of information content, the Cumulative Degrees of Freedom for Signal (CDFS) for temperature, defined as the vertical summation of DFS values from the surface up to the target altitude, is shown in Figure 1a. The results indicate a greater concentration of information content in the lowermost atmospheric layers for both AERI and HiSRAMS. Furthermore, most of the information content in temperature resides within the tropospheric region, specifically below 8 km, exhibiting DFS values of 6.14 and 3.41 for AERI and HiSRAMS, respectively.

The uncertainty associated with temperature retrievals varies between AERI and HiSRAMS. In AERI retrieval, the temperature uncertainty demonstrates an overall increase with altitude. In contrast, the temperature uncertainty in HiSRAMS retrieval decreases with altitude within the few initial levels near the surface. The interplay between the a priori uncertainty and the measurement uncertainty, the two terms in Eq. 7 that determine the posterior error covariance matrix, govern the retrieval accuracy. Within the troposphere, the a priori uncertainty typically decreases with altitude, thereby contributing to the overall reduction in retrieval uncertainty. The sensitivity of AERI measurements to temperature tends to decrease with altitude (Figure S5), leading to an increase in retrieval uncertainty with height (Figure 1a). At higher altitudes, this measurement uncertainty exceeds the a priori uncertainty. However, in the case of HiSRAMS measurements, the sensitivity to temperature does not show a monotonic decrease with altitude across certain channels, as shown in Figures S6a and S6b. This particular behavior results in maximum retrieval uncertainty within the lowermost atmospheric layers (Figure 1b). It is pertinent to note that the behavior of the sensitivity of the HiSRAMS measurements discussed above is influenced by variations in the thickness of vertical layers.



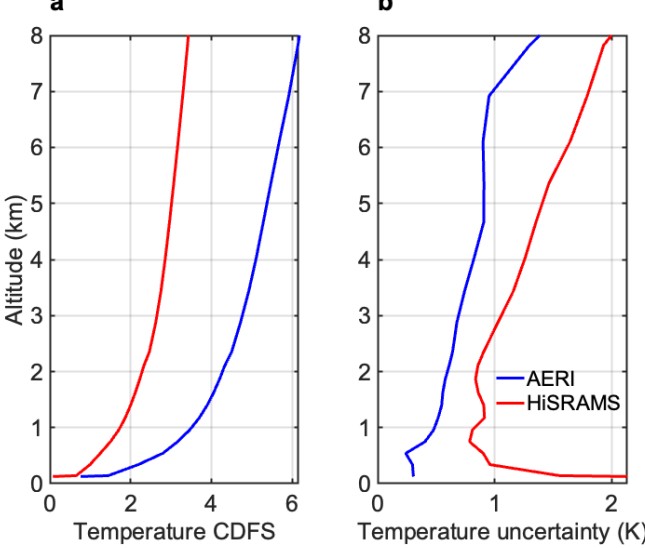


**Figure 1: Information content and retrieval uncertainty in temperature based on single ground measurements. (a) Cumulative Degrees of Freedom of Signal (CDFS) for temperature. (b) Uncertainty in the retrieved temperature.**

The validation of retrieved temperature profiles is based on ground-truth data from radiosonde measurements, represented by the black line in Figure 2. A temperature inversion at ~550 m, with a depth of approximately 400 m, presented a valuable

test to assess the resolvability of such signals in temperature retrieval. The subset displayed in Figure 2 specifically focuses on the profiles below 2 km. The a priori profile shows a near-surface temperature inversion (grey dashed line in Figure 2), which is different from the truth observed by the radiosonde. The AERI-retrieved temperature profile (blue line in Figure 2) effectively captures the sub-kilometer temperature inversion in the layer of 300-700 m together with an accurate representation of the near-surface temperatures below 300 m. In contrast, HiSRAMS cannot capture the detailed vertical

temperature structures below 2 km. The shape of the HiSRAMS-retrieved temperature profile (red line in Figure 2) closely mirrors that of the a priori profile. This discrepancy in the near-surface temperature feature retrievability between the two instruments can be attributed to AERI's higher SNR for temperature near the surface when compared to HiSRAMS, as demonstrated in Figures 3a, 3b, and 3c. In the upper troposphere, both AERI- and HiSRAMS-retrieved temperature profiles align well with the truth data. However, it is noted that AERI exhibits a more pronounced temperature retrieval bias above 6

km.

In summary, in this clear-sky ground-deployment case, it is evident that AERI outperforms HiSRAMS in terms of the retrievability of temperature profiles, showcasing superior performance across key metrics, including information content, retrieval uncertainty, and retrieval accuracy.





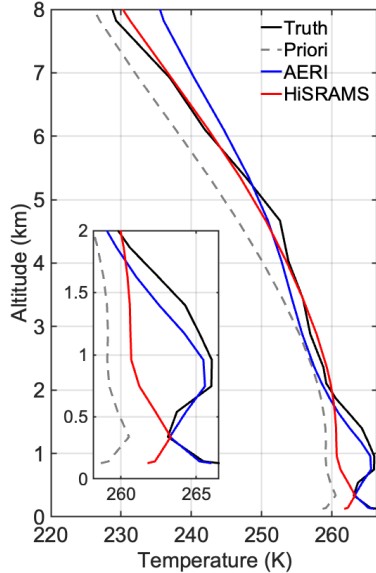


**Figure 2: Comparison between retrieved temperature profiles based on AERI or HiSRAMS ground measurement and the truth from radiosonde measurements.**

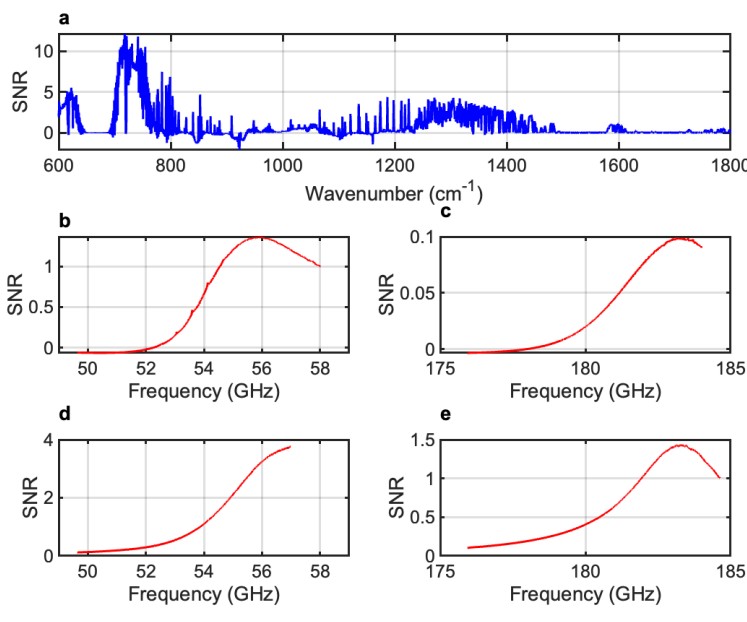

**Figure 3: Signal-to-Noise Ratio (SNR) for temperature. (a) SNR for AERI zenith-pointing measurements at 0.75 km (level 5). (b)**
**SNR for HiSRAMS zenith-pointing measurements in the oxygen band at 0.75 km (level 5). (c) SNR for HiSRAMS zenith-pointing measurements in the water vapor band at 0.75 km (level 5). (d) SNR for HiSRAMS nadir-pointing measurements in the oxygen band at 6.1 km (level 18). (e) SNR for HiSRAMS nadir-pointing measurements in the water vapor band at 6.1 km (level 18). Units: 1.**



## 4.2 Water vapor retrieval

The retrieval of atmospheric water vapor concentration using AERI and HiSRAMS ground measurements provides valuable insights. AERI records a total DFS of 4.22, whereas HiSRAMS reports 3.03, indicating that the two instruments offer comparable information regarding water vapor. However, at an altitude of 8 km, AERI reaches its maximum CDFS, while the CDFS for HiSRAMS continues to increase with altitude, suggesting that, despite its lower total water vapor DFS compared to AERI, HiSRAMS captures water vapor information over a broader vertical range.

Moreover, the uncertainty associated with retrieved water vapor concentration from AERI increases with altitude. In contrast, the uncertainty in HiSRAMS-retrieved water vapor concentration reaches a maximum of approximately 4 km. Notably, at altitudes below 5.5 km, AERI-retrieved water vapor exhibits lower uncertainties compared to HiSRAMS-retrieved water vapor, while the converse is observed at altitudes above 5.5 km.

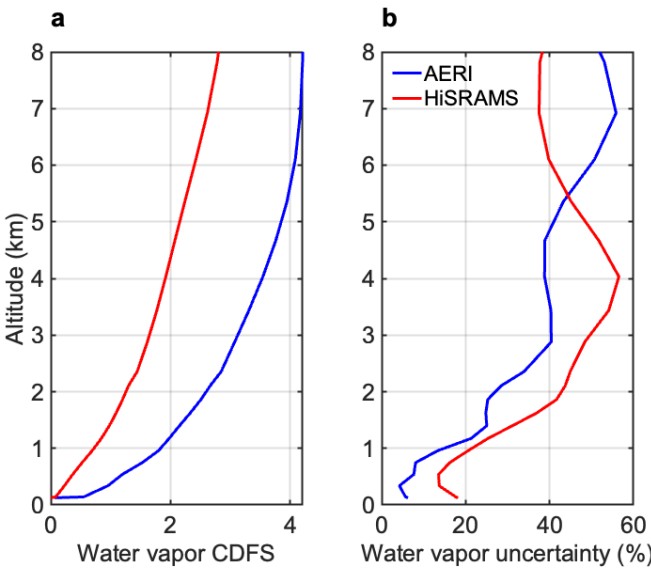

**Figure 4: Information content and uncertainty in water vapor retrieval from single ground-based measurements: (a) Cumulative Degrees of Freedom of Signal (CDFS) for water vapor, (b) Uncertainty in retrieved water vapor concentration.**

The water vapor concentration derived from radiosonde measurements (black line in Figure 5) reveals local minima and maxima, particularly a distinctive dry layer at approximately 750 m, with a depth of ~400 m. However, neither AERI nor HiSRAMS can fully capture this fine-scale, half-kilometer deep dry anomaly near the surface. The water vapor concentration

profile retrieved by AERI (blue line in Figure 5) exhibits a closer alignment with the radiosonde-derived truth, in contrast to the profile retrieved by HiSRAMS (red line in Figure 5). Furthermore, the AERI-retrieved water vapor concentration profile suggests the presence of a moist anomaly at an altitude of about 2 km, which is consistent with the radiosonde measurements. This finding, along with the resolvability of the temperature inversion shown above, encourages further investigation of the instruments' capacity to capture fine-scale thermodynamic variability in the lower atmosphere.




Generally, AERI outperforms HiSRAMS in water vapor retrieval, primarily due to the greater number of AERI channels with relatively higher SNR, as evident in Figures 6a, 6b, and 6c. The higher SNR in AERI measurements makes it more feasible to retrieve precise water vapor concentrations, particularly under challenging atmospheric conditions. However, both AERI and HiSRAMS exhibit lower water vapor retrievability than temperature retrievability, particularly in terms of information content and the ability to resolve confined sub-kilometer features.

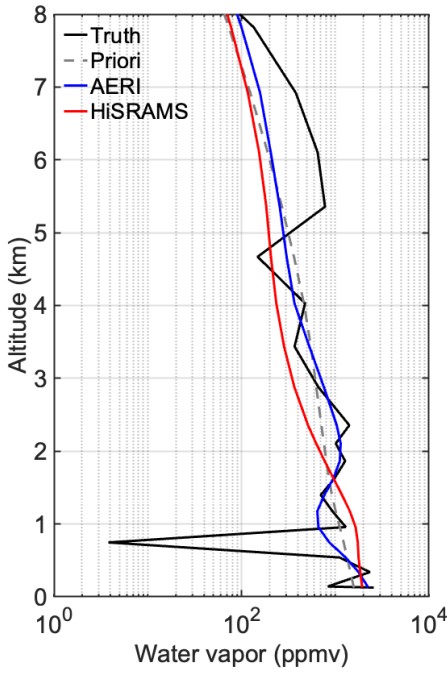


**Figure 5: Comparing retrieved water vapor concentration profiles from ground-based AERI and HiSRAMS measurements with radiosonde-derived truth.**



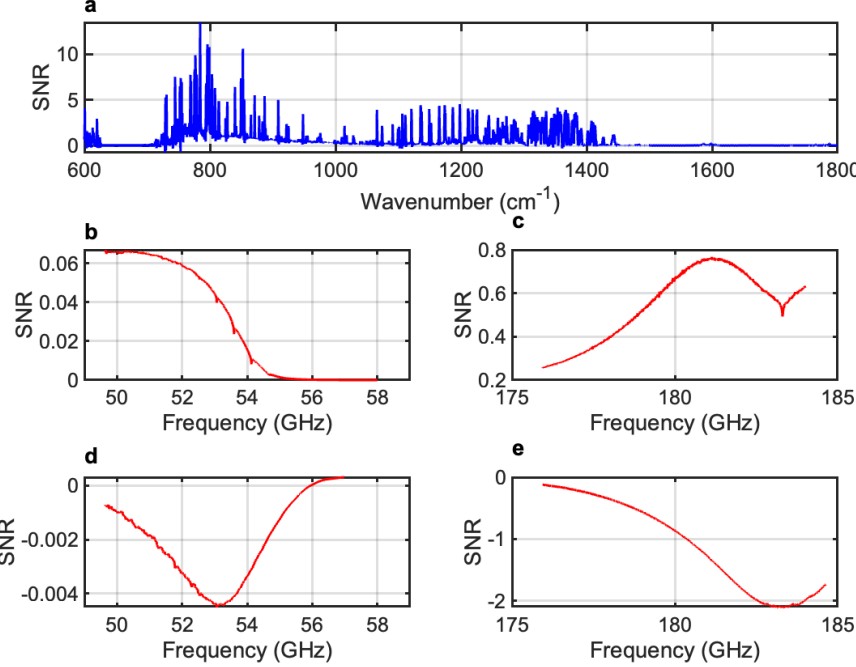

**Figure 6: Same as Figure 4 but for water vapor. Unit: 1. The presence of negative SNR values results from negative values in *K*.**

## 4.3 Sub-kilometer feature resolvability

Fortunately, in this specific case, both the temperature and water vapor vertical profiles exhibited sub-kilometer features, offering a valuable opportunity to assess the vertical resolvability of temperature and water vapor utilizing ground-based AERI and HiSRAMS measurements. A temperature inversion, with an altitude of approximately 550 m and a depth of 400 m, and a sudden dry layer at around 750 m with a depth of 400 m, were clearly discernible.

To understand the performance of the two instruments, we employ the averaging kernel matrix to determine the vertical resolution of the retrieved profiles. Each row in the matrix defines the averaging kernel for a specific level within the retrieved profile. Each row of an ideal *A* would look like a delta-function, indicating that the retrieved quantity at the vertical level exclusively represents the condition at that particular vertical location, i.e., exhibiting the highest attainable vertical resolution. However, due to correlations between different atmospheric layers, the vertical resolution of the retrieved profiles is constrained. Typically, each row of *A* reaches its peak at the retrieval level; this indicates that the bulk of information at that particular level is derived from that level, although smaller but non-negligible contributions are obtained from neighboring levels. Hence, we use the full-width half maximum (FWHM) of every row of *A* to quantify and represent the vertical resolution of the retrieval.



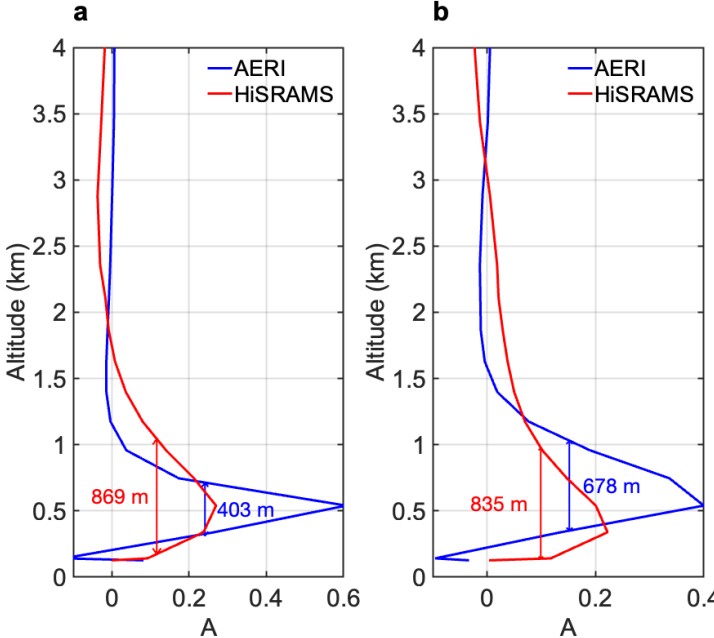

**Figure 7: Comparing vertical resolution of retrieved (a) temperature and (b) water vapor concentration profiles from ground-based AERI (blue lines) and HiSRAMS (red lines) measurements. This figure displays the averaging kernel matrix row for a vertical level at 550 m for temperature and 750 m for water vapor concentration.**

To demonstrate the concept, we show the rows of $A$ corresponding to altitudes of 550 m and 750 m, where the sub-kilometer temperature and water vapor concentration features are situated. For example, the blue line in Figure 7a shows the row of $A$ for AERI temperature retrieval at an altitude of 550 m. This row of $A$ peaks at the selected altitude, decreasing rapidly on either side of the peak. The AERI temperature retrieval at 550 m displays a FWHM of approximately 403 m, indicating its vertical resolution. In contrast, the HiSRAMS temperature retrieval at the same altitude offers a coarser vertical resolution of 869 m. The temperature inversion depth, roughly 400 m, closely aligns with AERI's vertical resolution but is shallower than that of HiSRAMS. This difference elucidates why the AERI retrieval can resolve the temperature inversion while the HiSRAMS retrieval cannot.

The vertical resolutions of the retrieved water vapor concentrations by AERI and HiSRAMS at 750 m are 678 m and 835 m, respectively. It is evident that neither of the rows in the averaging kernels of AERI nor HiSRAMS exhibit peaks at the target altitude, indicating a stronger correlation in water vapor concentrations between adjacent atmospheric layers. In general, AERI demonstrates higher water vapor vertical resolvability compared to HiSRAMS. However, the vertical resolutions for water vapor concentration at 750 m, as achieved by both AERI and HiSRAMS, exceed the depth of the thin dry layer, rendering them incapable of resolving this layer near the surface.



**4.4 Retrieval bias and uncertainty comparison**

The $3\sigma$ retrieval uncertainties (dashed lines in Figure 8), obtained from the posterior matrix (Eq. 7), are compared with the
retrieval bias (solid lines in Figure 8), which quantifies the difference between the retrieved profile and the truth derived
from radiosonde measurements. In general, both the retrieval bias in temperature and water vapor concentration falls within
the $3\sigma$ retrieval uncertainties, with exceptions at altitudes where fine vertical features exist.

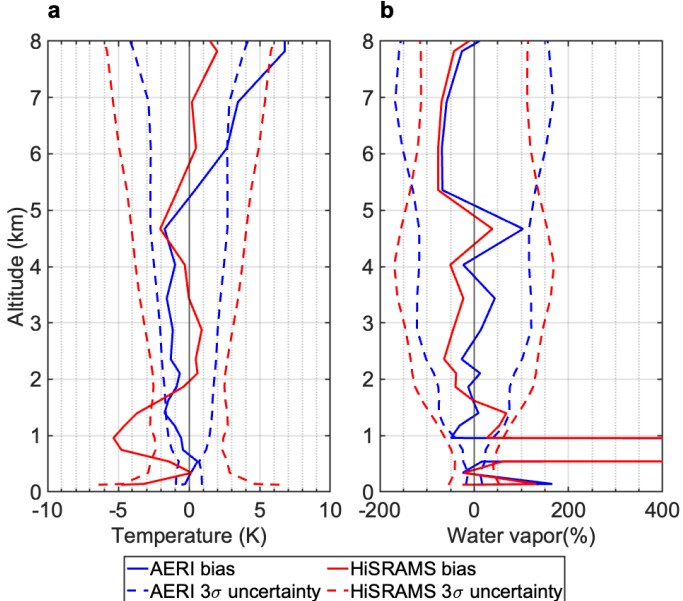

**Figure 8: Comparison of retrieval bias and uncertainty in ground-based retrievals for (a) temperature and (b) water vapor.**

The DFS of temperature and water vapor concentration in the troposphere is similar to previous AERI retrieval and
multichannel microwave radiometer retrieval studies (Blumberg et al., 2015; Loveless et al., 2022; Turner and Löhnert,
2021). Blumberg et al. (2015) conducted a comparison of retrieval performance between AERI and a 14-channel ground-
based microwave radiometer in clear-sky conditions. Even with a significantly greater number of channels in HiSRAMS
compared to the 14-channel radiometer, this study shows comparable retrieval performance between the two microwave
radiometers. This implies that, for ground-based retrievals, an increased number of channels in the microwave spectral range
may not necessarily improve the retrievals of clear-sky temperature and water vapor concentration.

In summary, when retrieving temperature and water vapor concentration profiles from hyperspectral ground measurements
under clear-sky conditions, infrared instruments exhibit better performance compared to microwave instruments in terms of
information content, retrieval uncertainty, vertical feature resolvability, and retrieval biases. However, it is worth noting that
hyperspectral microwave instruments demonstrate less temperature bias and lower water vapor concentration uncertainty in
the upper troposphere.



## 5 Joint airborne HiSRAMS and ground-based AERI retrievals

As an airborne instrument, HiSRAMS was used to collect radiance measurements at various altitudes during the FC2023 campaign. To investigate the potential advantages of combining these airborne HiSRAMS measurements with ground-based AERI measurements, we conducted joint retrievals in comparison to independent retrievals. We refer the measurements obtained from this unique setting as "sandwich" measurements. Throughout FC2023, we gathered HiSRAMS nadir-pointing measurements during ten flight legs, covering altitudes from near the surface up to 6.8 km. We specifically selected measurements obtained at 6.8 km for our joint retrievals, as the measurements at this altitude captured a substantial portion of the troposphere viewed by both instruments.

In this section, we compare joint retrievals, which combine the AERI zenith-pointing measurements on the ground and the HiSRAMS nadir-pointing measurements at 6.8 km to retrieve temperature and water vapor concentration vertical profiles, with single retrievals based on either the AERI zenith-pointing measurements on the ground or the HiSRAMS nadir-pointing measurements at 6.8 km alone. In the case of joint retrieval and AERI-only retrieval, $n_{level}$ is set at 76, as ground-based measurements are incorporated into both retrievals. To avoid uncertainties due to land surface emissivity, we adopt an elevated "surface" boundary condition at altitude of 429 m for the HiSRAMS nadir-pointing forward model (Liu et al., 2023). Consequently, for HiSRAMS nadir-pointing retrievals alone, $n_{level}$ is set at 30, considering that there are only 15 levels between 429 m and 6.8 km in the vertical configuration used for these retrievals. Previous work by Liu et al. (2023) has identified biases in nadir-pointing HiSRAMS flight measurements. To ensure the reliability of our retrieval results, we have corrected the HiSRAMS nadir-pointing measurements used in this study, following the method outlined in Appendix A.

### 5.1 Temperature retrievals

Joint retrieval enhances the temperature information content. The total temperature DFS for the joint retrieval stands at 10.96, surpassing the values obtained in individual retrievals (9.52 for AERI-only retrieval and 3.20 for HiSRAMS-only retrieval). A field campaign carried out by the UK Met Office Airborne Research Interferometer Evaluation System (ARIES) demonstrated a DFS between 4 and 5 for temperature retrievals using airborne nadir-pointing infrared hyperspectral observations between 690 and 775 cm⁻¹ at similar observational height to HiSRAMS (Allen et al., 2014). This DFS value is higher than that of HiSRAMS retrieval.

Figure 9a shows the detailed DFS values for specific altitude levels. In the case of the HiSRAMS-only nadir-pointing retrieval, the DFS increases with altitude (red line in Figure 9a). This increase is attributed to the HiSRAMS measurements acquired at 6.8 km, making it more responsive to atmospheric conditions near the instrument. Similarly, for the AERI-only zenith-pointing retrieval, the DFS decreases with altitude (blue line in Figure 9a). Notably, the DFS for the joint retrieval (green line in Figure 9a) exceeds that of either individually, signifying the increased information content, particularly in the upper troposphere, where AERI's capabilities are limited.



Joint retrieval not only enhances the information content but also diminishes retrieval uncertainty. The temperature
uncertainty in AERI or HiSRAMS single-instrument retrievals increases and decreases with altitude respectively. By
combining both sets of measurements, the overall uncertainty in temperature diminishes, compared to that in either of the
individual retrievals. Consequently, temperature uncertainties below 6 km consistently remain within 1 K.

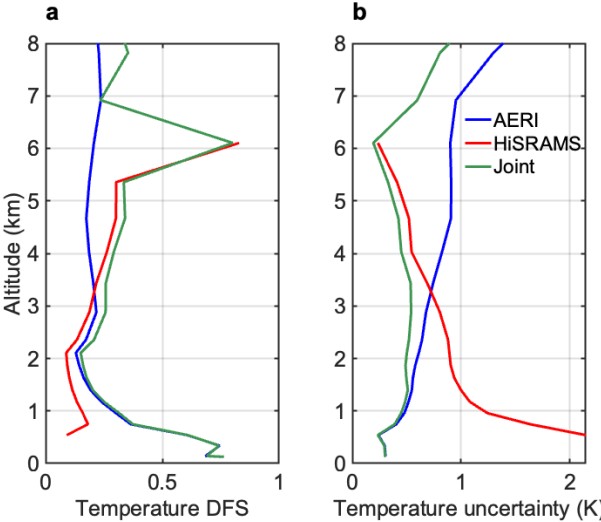

**Figure 9: Comparison of information content and temperature retrieval uncertainty between joint airborne HiSRAMS and**
**ground-based AERI retrievals versus single-instrument retrievals from either airborne HiSRAMS or ground-based AERI. (a)**
**Temperature Degrees of Freedom for Signal (DFS). (b) Uncertainty in retrieved temperature.**

Figure 10a shows the temperature profiles retrieved from actual measurements. As with the ground-based HiSRAMS
retrieval results, the HiSRAMS-only nadir-pointing retrieval (red line in Figure 10a) remains incapable of resolving the fine
vertical temperature features near the surface. This limitation arises from a lower SNR (not shown), compared to the zenith-
pointing HiSRAMS measurements, in regions where a temperature inversion is present. The smaller SNR results from the
measurements being further away from the fine vertical feature. The change in observation locations identifies why the
temperature profile retrieved from HiSRAMS-only airborne measurements around 6 km closely approximates the actual
values, corroborated by the relatively higher SNR shown in Figure 3d.

As discussed in the previous section on ground-based retrieval, the AERI-only retrieval successfully resolves temperature
inversions near the surface but exhibits a larger temperature bias in the upper troposphere. Joint retrieval combines the
strengths of the two instruments by  resolving the fine vertical features near the surface and yielding a reduced temperature
bias in the upper troposphere compared to the AERI-only retrieval. Note that a temperature bias still exists above 6 km, even
in the case of joint retrieval.





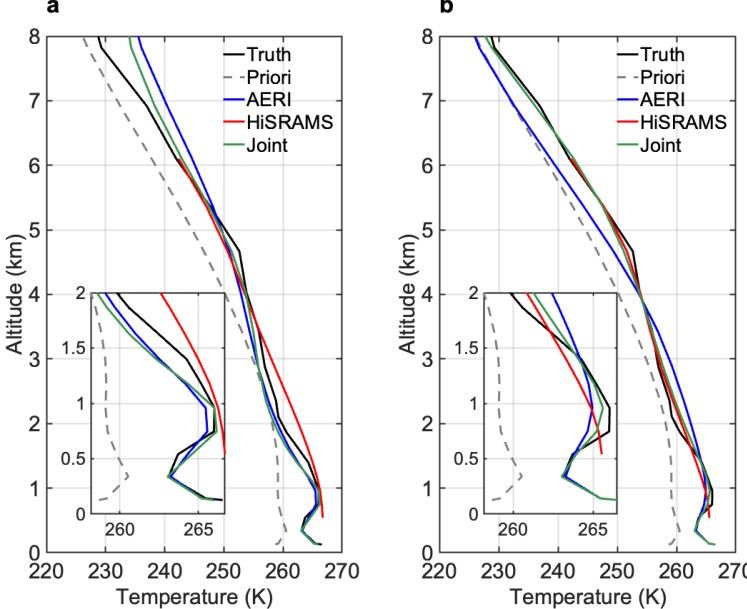

**Figure 10: Retrieved temperature profiles for joint retrieval (combining AERI zenith-pointing measurements on the ground and HiSRAMS nadir-pointing measurements at 6.8 km) and single-instrument retrievals (utilizing either AERI zenith-pointing measurements on the ground or HiSRAMS nadir-pointing measurements at 6.8 km). (a) Retrieved temperature profiles based on actual measurements. (b) Retrieved temperature profiles based on synthetic measurements.**

Liu et al. (2023) identified a significant brightness temperature bias in HiSRAMS airborne measurements concerning the brightness temperature truth derived from the HiSRAMS forward model, which utilized radiosonde measurements as inputs. This bias may arise from inaccuracies in either the HiSRAMS measurements or the brightness temperature truth due to imprecise atmospheric state inputs. To assess the limits of the joint retrieval concept, we further conducted joint and single-instrument retrievals based on synthetic measurements. Specifically, radiosonde-derived temperature and water vapor profiles served as inputs for AERI and HiSRAMS forward models to generate synthetic spectra with added random noises appropriate to the measurement uncertainty to emulate the measurements in the retrieval algorithm.

The AERI retrieval based on the synthetic measurement exhibited good resolvability of the temperature inversion near the surface but displayed a larger bias in upper tropospheric temperature (blue line in Figure 10b). Simultaneously, the synthetic HiSRAMS retrieval accurately captured temperature profiles well below the observational altitude but could not resolve the near-surface temperature inversion feature (red line in Figure 10b). In contrast, the joint synthetic retrieval not only captured the near-surface temperature inversion feature but also effectively constrained the temperature profile both above and below the observational altitude (green line in Figure 10b). This underscores the complementary information in the two instruments and a substantial potential of joint retrieval between AERI and HiSRAMS.



## 5.2 Water vapor retrievals

Joint retrieval increases the information content in water vapor concentration. The total water vapor DFS for joint retrieval is 5.82, exceeding that of the water vapor DFS values for either AERI- or HiSRAMS-only retrievals, which are 4.22 and 3.11, respectively. The water vapor DFS for specific levels, illustrated in Figure 11a, clearly indicates the enhanced water vapor information content achieved through joint retrieval. Joint retrieval in water vapor particularly excels at the HiSRAMS observation altitude and near-surface, primarily contributed by HiSRAMS and AERI, respectively. Allen et al. (2014) reported a DFS of approximately 3 for the ARIES airborne system at an observational height of 7.4 km with 10 vertical levels, comparable to the HiSRAMS retrieval result. Similarly, higher information content was detected closer to the observational height.

Joint retrievals reduce uncertainties in the retrieved water vapor concentration. As with joint temperature retrievals, uncertainties in retrieved water vapor concentrations from AERI-only measurements generally decrease with altitude. Notably, uncertainties in retrieved water vapor concentration profiles from HiSRAMS-only measurements exhibit a distinct peak around 2 km. Water vapor concentration profiles computed from joint retrievals are characterized by reduced uncertainties at all levels, including those above the HiSRAMS observational altitude.

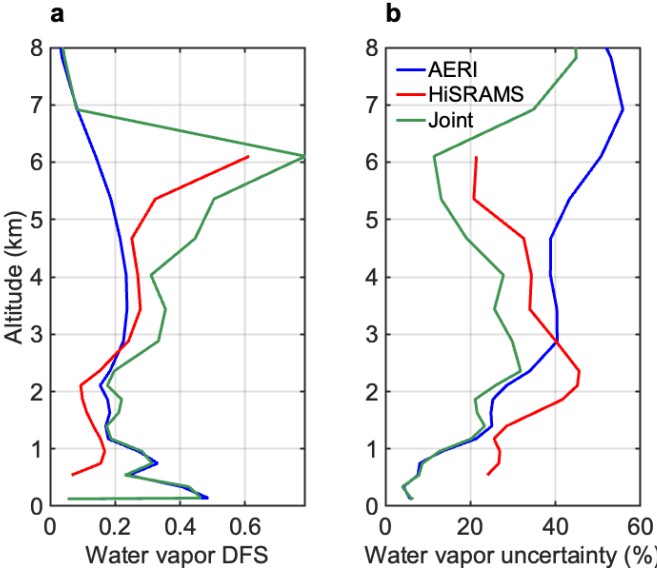

**Figure 11: Comparison of information content and uncertainty in water vapor retrievals between joint airborne HiSRAMS and ground-based AERI retrievals and single-instrument retrievals from either airborne HiSRAMS or ground-based AERI. (a) Water vapor Degrees of Freedom for Signal (DFS). (b) Uncertainty in retrieved water vapor concentration.**

Water vapor concentration profiles retrieved from actual measurements are presented in Figure 12a. While a HiSRAMS-only retrieval constrains water vapor concentration near the observation altitude due to some channels with the absolute value of SNR larger than 1 (Figure 6e), its retrieval capability diminishes further away from it. Joint retrieval in water vapor



concentration combines the strengths of both AERI- and HiSRAMS-only retrievals. Nevertheless, constrained by vertical
resolution limitations, even joint retrieval falls short of fully capturing fine vertical water vapor features, such as thin dry
layers around 750 m and 4.6 km.

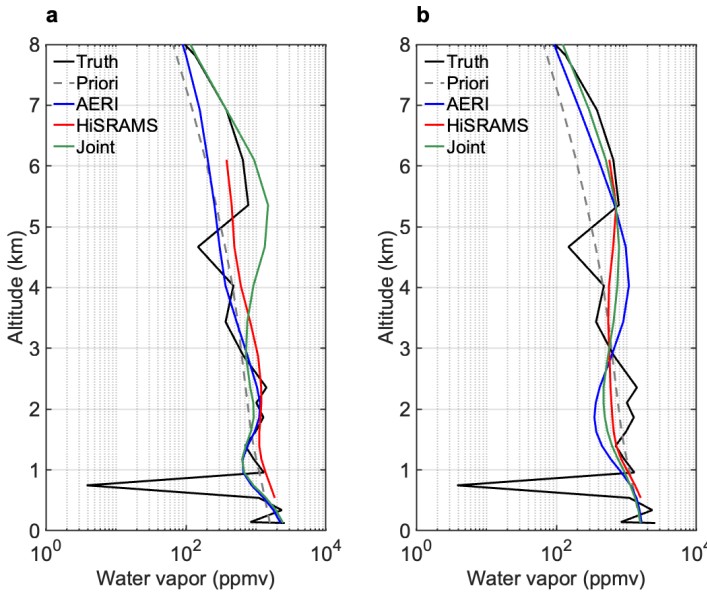

**Figure 12: Retrieved water vapor concentration profiles for joint retrieval (combining both AERI zenith-pointing measurements on the ground and HiSRAMS nadir-pointing measurements at 6.8 km) and single-instrument retrievals (utilizing either AERI**
**zenith-pointing measurements on the ground or HiSRAMS nadir-pointing measurements at 6.8 km). (a) Retrieved water vapor profiles based on actual measurements. (b) Retrieved water vapor profiles based on synthetic measurements.**

The synthetic retrieval results in Figure 12b do not exhibit significant improvement in terms of retrieved water vapor
concentration bias and the resolution of fine vertical features. This suggests that the accuracy in water vapor retrieval is not
severely limited by the accuracy of radiance measurements.

**5.3 Retrieval bias and uncertainty comparison**

Generally, the retrieval bias is within the $3\sigma$ retrieval uncertainties for HiSRAMS nadir-pointing flight measurements single-
instrument retrieval and joint retrieval (Figure 13). Significant retrieval biases exist at altitudes with fine vertical features.
The temperature and water vapor retrieval uncertainties decrease with distances away from the observational heights in
single instrument retrievals. Joint retrieval shows a bow-shaped posterior uncertainty, signaling its benefits in reducing the
retrieval uncertainty. The retrieval bias reported here was analyzed from only one case study, which affords limited error
statistics.



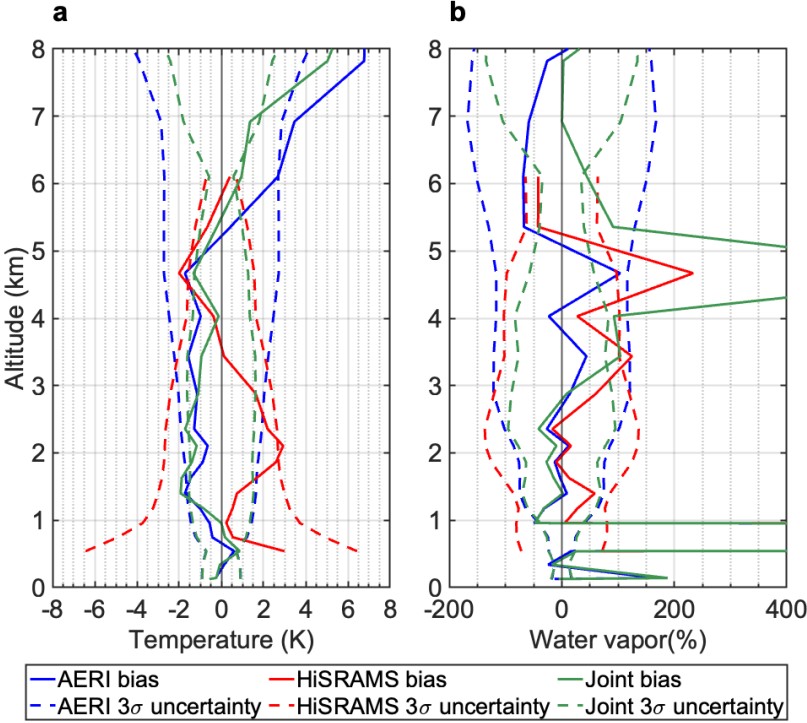

**Figure 13: Comparison of retrieval bias and uncertainty in joint retrievals: (a) Temperature, (b) Water vapor.**

In summary, the synergistic integration of radiative measurements at varying observational altitudes enhances the retrieval performance of thermodynamic variables, with a notable impact on temperature. Previous studies have demonstrated similar improvements, such as the synergy achieved through hyperspectral infrared instruments at different altitudes (e.g., Loveless et al., 2022; Zhao et al., 2022; Shahabadi and Huang, 2014). This study highlights the potential of synergistic retrievals, specifically emphasizing the advantages achieved by combining hyperspectral infrared and microwave radiometers across different altitudes, which may be especially useful for intensive observation campaigns.

## 6 Conclusions and discussion

Hyperspectral radiance measurements afford an advantageous means to monitor the vertical distributions of temperature and water vapor concentration. Leveraging advancements in polyphase spectrometers, a hyperspectral microwave radiometer, featuring a large number of spectral channels comparable to hyperspectral infrared radiometers has been developed. In this study, measurements from an airborne hyperspectral microwave radiometer, HiSRAMS, and a ground-based hyperspectral infrared radiometer, AERI, were acquired on February 11, 2023, to test their retrievals of temperature and water vapor vertical profiles.



We first evaluated the retrieval performance of ground-based AERI measurements against ground-based HiSRAMS measurements. Concerning retrieval uncertainty and information content, AERI demonstrates superior retrieval performance for both temperature and water vapor concentrations compared to HiSRAMS, except in the water vapor retrieval at higher
altitudes above 5.5 km. Both AERI and HiSRAMS retrievals exhibit a higher information content for temperature than for water vapor concentrations. The high vertical retrieval resolution of AERI enables the resolution of fine temperature inversion features near the surface, a capability not shared by HiSRAMS temperature retrieval. On the other hand, neither AERI nor HiSRAMS can resolve fine vertical features of water vapor, such as the thin dry layers found near the surface, due to the coarse vertical retrieval resolution of their retrievals. While AERI captures the overall water vapor profile effectively,
HiSRAMS demonstrates reduced retrieval performance in this application. These results suggest that, for ground-based measurements, an increase in the number of channels of microwave radiometers does not necessarily make them comparable to infrared hyperspectral radiometers.

We also experimented with a joint retrieval approach involving ground-based zenith-pointing AERI measurements and airborne nadir-pointing HiSRAMS measurements. We find this approach enhances the performance of temperature and
water vapor concentration retrievals compared to single instrument retrievals. The joint retrieval exhibits increased information content and reduced retrieval uncertainty for temperature and water vapor concentrations across all retrieval levels. Ground-based AERI measurements contribute to the resolution of near-surface temperature features, while airborne HiSRAMS measurements exhibit a lower retrieval bias in temperature near the observational altitude (6.8 km). Combining measurements from both instruments yields retrieved temperature profile that captures fine vertical features near the surface
while mitigating bias in upper-tropospheric temperatures near the HiSRAMS observational altitude. By comparison, the improvement in accuracy in the water vapor concentration retrieval is limited in joint retrieval.

This study is subject to certain limitations. Using the entire spectrum of channels for both instruments in conducting temperature and water vapor retrievals ensures maximum information content employed in the retrievals but is also subject to larger errors and possible interference in certain channels, which can be minimized or eliminated via a channel selection
approach in future work. Additionally, the retrieval comparison in this study relies on limited samples from a single campaign, thus bounding the usefulness of the error statistics and comprehensiveness of this assessment. This issue, also relegated to future work, can be addressed with more field observations.

In conclusion, this study utilizes infrared and microwave hyperspectral radiometers to retrieve clear-sky temperature and water vapor concentration profiles under various observational conditions. The retrieval comparison between HiSRAMS and
AERI ground-based measurements reveals that infrared hyperspectral observations provide a higher information content and greater vertical resolution for temperature and water vapor retrievals than microwave hyperspectral observations. However, employing zenith-pointing AERI measurements and nadir-pointing HiSRAMS measurements at high altitudes, forming a "sandwich" configuration, not only enhances information content but also reduces retrieval uncertainty and bias in temperature and water vapor concentrations. Integrating ground-based infrared and airborne microwave hyperspectrometers
proves advantageous for sounding temperature and water vapor profiles. To thoroughly assess and explore the potential of

hyperspectral microwave radiometers in retrieving thermodynamic profiles, further case studies addressing both clear-sky and cloudy-sky temperature and water vapor retrievals are warranted.

**Data Availability**

The HiSRAMS and AERI measurements data are well described and can be obtained from Liu et al. (2023). The selected matrices in retrievals used in this study, along with the retrieved temperature and water vapor profiles, can be obtained from the Mendeley Data (http://doi.org/10.17632/524hj3w6r8).

**Author contribution**

YH, NB, JG, and MW conceived the research. NB, LL, SX and YH conducted the field data collection. LL developed the AERI retrieval algorithm. NB and PG, with modifications provided by LL, developed the HiSRAMS retrieval algorithm. Furthermore, LL developed the joint retrieval algorithm. LL led the manuscript writing process, with contributions from all co-authors.

**Competing interests**

The authors declare that they have no conflict of interest.

**Acknowledgments**

We extend our gratitude to our colleagues from the Atmospheric Radiation Group at McGill University for their valuable comments on this manuscript. We thank Jon Gero and David Turner, and Olivier Auriacombe, for their advice on the analysis of AERI and HiSRAMS measurements, respectively. The successful completion of the measurement campaign was made possible through the contributions of numerous individuals from NRC, McGill University, and Omnisys Instruments. In particular, the flight data collection was supported by NRC scientists Dr. Cuong Nguyen, Dr. Leonid Nichman, Dr. Keyvan Ranjbar, Mr. Kenny Bala and others in Airborne Facilities for Atmospheric Research and Reconnaissance (AFARR) team; the AERI and ground facility operation were supported by Eve Bigras, Calin Giurgiu, Véronique Meunier and other McGill staff. This work was supported by grants from the Canadian Space Agency (19FAMCGB16 and 21SUASATHC) and the Fonds de Recherche du Québec Nature et technologies (PR-283823). Additionally, we acknowledge the funding provided by the European Space Agency (ESA contract 4000123417/NL/LA) for HiSRAMS and the support for AERI from the Canada Foundation for Innovation (CFI-36146), the Quebec Government, McGill University, and Université du Québec à Montréal.



**Appendix A: Bias correction for HiSRAMS nadir-pointing measurements at 6.8km**

Nadir-pointing HiSRAMS measurements exhibit some brightness temperature biases (Liu et al., 2023), which need to be removed for accurate physical retrieval applications. Given that our focus is solely on nadir-pointing measurements during a
specific leg, and we obtain true temperature and water vapor concentration profiles from radiosonde measurements, the brightness temperature bias can be identified based on the differences between HiSRAMS measurements and forward model simulations (blue lines in Figure A1).

We partitioned the entire spectrum into distinct spectral ranges, each defined by specific bias features. Within each spectral range, we determined the brightness temperature bias either as a constant or through the application of a linear regression
method, as illustrated by the red lines in Figure A1. Subsequently, the biases represented by the red lines were systematically removed for all retrieval cases utilizing nadir-pointing HiSRAMS measurements at 6.8 km.

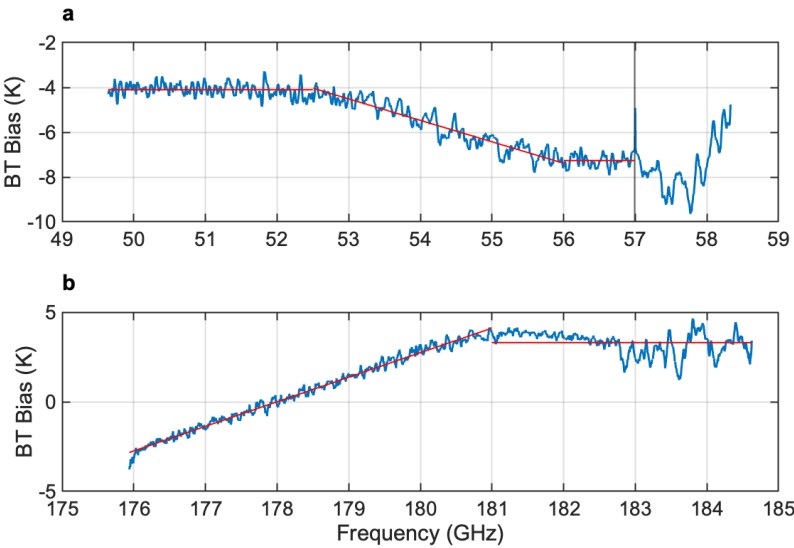

**Figure A1. Correction of nadir-pointing HiSRAMS measurements bias in (a) oxygen band and (b) water vapor band. The blue lines represent the brightness temperature bias determined by the difference between measurements and forward model**
**simulations (refer to Liu et al., 2023 for detailed methodology). The red lines represent the determined bias.**

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
