# Peer review of "Comparative experimental validation of microwave hyperspectral atmospheric soundings in clear-sky conditions"

_EGUsphere, 2024_

## Referee Comment (RC1)

**Review of the preprint AMT manuscript egusphere-2024-1045:**

**Comparative experimental validation of microwave hyperspectral atmospheric soundings in clear-sky conditions**

General comment:

This is an excellent paper which describes a genuine scientific experiment using collocated hyperspectral infrared and microwave atmospheric soundings for retrieval of its thermodynamic states. The approach and methodologies applied to the experiment appear sound, and the presentation of the retrieval results is clear. There is some lack of clarity/rigor in the presentation details of the experiment methodologies and of the measurement parameters/conditions. Those aspects are commented below for improving further the quality of the paper. Minor editorial comments are listed at the end.

Comments for improving the lack of clarity/rigor in the presentation details of the experiment methodologies and of the measurement parameters/conditions:

Lines 72 – 82:  2 Field campaigns

It would be good, for the readability, to summarise the field campaigns in a table format together with relevant information such as the dates, locations, experiment configurations (geometry & polarization), wind condition, additional in-situ measurements, etc.

Lines 83 – 133:  Figures S1, S2, …, S7

Figures S1, S2, .., S7 are mentioned in the text, but are nowhere to be found. It is not understandable whether they are referring to those of a specific reference paper. Please clarify!

Lines 115 – 121:  ERA5 re-analysis dataset

The description of the re-analysis dataset used as the a priori dataset should be improved. For instance, why is it necessary to have a historical Februaries dataset from 1944 to 2022? Do the 9 grid boxes correspond to 3x3 boxes with the center box containing the Ottawa Airport? In which of the boxes is the campaign instrumentation located? A map with the campaign location, ERA5 grids and flight path would be helpful for the readers.

Line 170:  Eq. 9

As $K$ can be negative (Figs. 6d & 6e), wouldn't it better to use $|K|$ in Eq. 9?

$S_{e,diag}$ can in general be improved by averaging of independent measurement samples, limited by the Allan variance of the instrumentation. Please explain how the instrumentation data have been averaged for achieving the optimum $S_{e,diag}$.

Lines 213 – 214:  Radiosonde data

How well are the collocation of the radiosonde measurements with respect to the AERI/HiSRAMS profile?

Lines 334 – 336:  "… we adopt an elevated surface boundary condition at altitude of 429 m …"

Please include a short explanation how the land surface emissivity is artificially reduced during the retrieval.

Lines 428 – 429: "The temperature and water … **increase** with distances away … in single instrument retrievals **due to the increasing atmospheric attenuation**.

Editorial comments:"

Figures 2, 5, 10 & 12: Please expand the horizontal axis for a better visualization of the different retrieved profiles.

Line 50: "However, recent advancements in **digital** Fast Fourier **Transform** (FFT) **spectrometer** techniques have led to the development of …"

Line 87: "The AERI forward model **which** we adopt is …

Line 118: "The vertical coordinate  adopted in …"

Line 224: "However, it is noted in **Figure 2** that AERI exhibits a more pronounced …"

Line 504: "…European Space Agency (ESA contract 4000123417/NL/LA) for **the** HiSRAMS **development and permission for its use,** and …"

---

## Author Comment (AC1)

In this response letter, our responses are labeled in blue. Revised sentences in the updated manuscript are shown in *italic purple text*. All line numbers correspond to the revised manuscript with tracked changes, where deletions are shown in red and additions are shown in blue.

General comment:

This is an excellent paper which describes a genuine scientific experiment using collocated hyperspectral infrared and microwave atmospheric soundings for retrieval of its thermodynamic states. The approach and methodologies applied to the experiment appear sound, and the presentation of the retrieval results is clear. There is some lack of clarity/rigor in the presentation details of the experiment methodologies and of the measurement parameters/conditions. Those aspects are commented below for improving further the quality of the paper. Minor editorial comments are listed at the end.

We thank Dr. Lin for the suggestions and comments, which greatly improved the clarity and quality of our manuscript.

Comments for improving the lack of clarity/rigor in the presentation details of the experiment methodologies and of the measurement parameters/conditions:

Lines 72 – 82: 2 Field campaigns

> It would be good, for the readability, to summarise the field campaigns in a table format together with relevant information such as the dates, locations, experiment configurations (geometry & polarization), wind condition, additional in-situ measurements, etc.

We have included a table describing the basic information of the field campaign, as well as the data collected and used in this retrieval analysis.

**Table 1. Summary of the field campaign.**

| Date | 11 February 2023 |
|---|---|
| Location | Ottawa International Airport (latitude: 45.32°, longitude: -75.66°), Ottawa, Canada |
| Radiosonde | One radiosonde sounding between 14:22:53-15:26:22 UTC. |
| HiSRAMS | HiSRAMS was mounted on a research aircraft, providing zenith-pointing measurements before take-off and after landing, and both zenith-pointing and nadir-pointing measurements at various altitudes. In this study, we primarily use single-polarized zenith-pointing measurements from the ground and nadir-pointing measurements taken at 6.8 km altitude. |
| AERI | AERI was positioned on the ground, providing continuous zenith-pointing downwelling longwave radiance measurements. |

Lines 83 – 133: Figures S1, S2, ..., S7

> Figures S1, S2, .., S7 are mentioned in the text, but are nowhere to be found. It is not understandable whether they are referring to those of a specific reference paper. Please clarify!

Figures S1 to S7 are provided in the supplemental document, which can be accessed at https://doi.org/10.5194/egusphere-2024-1045-supplement.

Lines 115 – 121: ERA5 re-analysis dataset

> The description of the re-analysis dataset used as the a priori dataset should be improved. For instance, why is it necessary to have a historical Februaries dataset from 1944 to 2022? Do the 9 grid boxes correspond to 3x3 boxes with the center box containing the Ottawa Airport? In which of the boxes is the campaign instrumentation located? A map with the campaign location, ERA5 grids and flight path would be helpful for the readers.

We aim to incorporate a variety of possibilities to form the *a priori* dataset. At the same time, we want to exclude the seasonal cycle from the *a priori* dataset. Therefore, we use the historical February dataset from 1944 to 2022, sourced from ERA5, to construct the *a priori* dataset. We have revised the sentence on Lines 133-134: *"This setup was designed to capture the temporal and spatial variability of atmospheric state variables near the measurement site."*.

Yes, the 9 grid boxes correspond to a 3x3 arrangement, with the center box containing Ottawa International Airport, where all instruments were located to collect the data used in the retrieval analysis. We have revised the sentences on Lines 92-94: *"A summary of the FC2023 field campaign is provided in Table 1, and all data used in the retrieval analysis were collected at Ottawa International Airport, Ottawa, Canada."* and on Lines 131-133: *"Hourly-mean profiles from Februarys between 1944 and 2022 were extracted from ERA5 across a 3x3 grid of nine boxes, with the center box including Ottawa International Airport (latitude: 45.32°, longitude: -75.66°), where real measurements were collected."*.

Line 170: Eq. 9

> As K can be negative (Figs. 6d & 6e), wouldn't it better to use |K| in Eq. 9?

We have revised the definition of **SNR** and have updated Figure 3 and Figure 6. We have included one sentence on Lines 206-208: *"To avoid negative **SNR** due to negative **K** values, which indicate how the measurements increase or decrease in response to increases in the state variables, we take the absolute values of **K** in the formula."*.

> $S_{e,diag}$ can in general be improved by averaging of independent measurement samples, limited by the Allan variance of the instrumentation. Please explain how the instrumentation data have been averaged for achieving the optimum $S_{e,diag}$.

For the AERI observations, $\mathbf{S_{e,diag}}$ is determined by the noise equivalent spectral radiance. Based on a simple sensitivity test, the information content for temperature and water vapor from AERI measurements is primarily limited by $\mathbf{K}$, rather than by $\mathbf{S_{e,diag}}$ (see Table R1-1). This indicates that averaging independent measurement samples will not significantly improve the retrieval performance of temperature and water vapor from AERI measurements.

**Table R1-1 The impact of $\mathbf{S_{e,diag}}$ on the information content using AERI measurements**

| $\mathbf{S_{e,diag}}$ scaling factor | DFS for temperature | DFS for water vapor |
|---|---|---|
| 0.2 | 10.82 | 4.83 |
| 1 | 9.52 | 4.22 |
| 5 | 8.32 | 3.58 |

For the HiSRAMS observations, the instrument and model uncertainty that form $\mathbf{S_{e,diag}}$ are defined in Liu et al., 2024: "For HiSRAMS measurements, if multiple individual measurements are averaged, the standard deviation of any individual measurements during the whole observational period is considered to be the uncertainty of the HiSRAMS-averaged measurements, which is applied to HiSRAMS ground measurements in FC2021 and FC2022 and flight measurements in FC2023. If only the individual observed spectrum is available, i.e., FC2023 HiSRAMS ground measurements, its uncertainty is determined by taking into account the radiometric noise characterized by the noise-equivalent differential temperature, calibration load imperfections, detector nonlinearity error, and instrument drift (Bliankinshtein et al., 2023a).". HiSRAMS sensor development took into account the trade-off between spectral resolution, temporal/spatial resolution of measurement and noise level, such that instrument integration time is compliant with Allan variance and reasonable for sensor on a moving platform, while keeping the hyperspectral resolution of 6.1 MHz. Sensitivity tests involving scaling (i.e., inflating or reducing) the matrix indicate that $\mathbf{S_e}$ is not the primary controlling factor of retrieval information content; instead, $\mathbf{K}$ is (Bliankinshtein et al., 2023).

Liu, L., Bliankinshtein, N., Huang, Y., Gyakum, J. R., Gabriel, P. M., Xu, S., and Wolde, M.: Radiative closure tests of collocated hyperspectral microwave and infrared radiometers, Atmospheric Measurement Techniques, 17, 2219-2233, 2024.

Bliankinshtein, N., Liu, L., Gabriel, P., Xu, S., Bala, K., Wolde, M., Huang, Y., Auriacombe, O., Krus, M., and Angevain, J.-C.: Airborne validation of HiSRAMS atmospheric soundings, IGARSS IEEE International Geoscience and Remote Sensing Symposium, 4372-4375, 2023.

Lines 213 – 214: Radiosonde data

How well are the collocation of the radiosonde measurements with respect to the AERI/HiSRAMS profile?

A single radiosonde was launched at Ottawa International Airport between 14:22:53 and 15:26:22 UTC. The balloon's trajectory is illustrated in Figure 2 from Liu et al. (2024). Since the radiosonde profile was obtained in slightly over one hour, the temporal variability of the profiles is expected to be minimal. However, the spatial variability of the profiles could lead to relatively large errors. We have included this discussion in the Conclusions and discussion section on Lines 516-517: *"Specifically, only a single radiosonde was launched during the field campaign, which may have induced temporal and spatial variability in the truth profile."*.

Liu, L., Bliankinshtein, N., Huang, Y., Gyakum, J. R., Gabriel, P. M., Xu, S., and Wolde, M.: Radiative closure tests of collocated hyperspectral microwave and infrared radiometers, Atmospheric Measurement Techniques, 17, 2219-2233, 2024.

Lines 334 – 336: "... we adopt an elevated surface boundary condition at altitude of 429 m ..."

Please include a short explanation how the land surface emissivity is artificially reduced during the retrieval.

We have included two sentences on Lines 375-377: *"Instead of developing a land surface emissivity model to simulate the surface boundary condition, we employ the HiSRAMS observed brightness temperature at 429 m as the "surface" (lower boundary condition). This approach accounts for both the surface emission and the atmospheric effects below 429 m."*.

Lines 428 – 429: "The temperature and water ...  **increase** with distances away ... in single instrument retrievals **due to the increasing atmospheric attenuation**.

Done.

Editorial comments:

Figures 2, 5, 10 & 12: Please expand the horizontal axis for a better visualization of the different retrieved profiles.

We have updated Figures 2, 5, 10, and 12.

Line 50: "However, recent advancements in  **digital** Fast Fourier  **Transform** (FFT)  **spectrometer** techniques have led to the development of ..."

Corrected.

Line 87: "The AERI forward model **which** we adopt is ...

Done.

Line 118: "The vertical coordinate  adopted in ..."

Done.

Line 224: "However, it is noted in **Figure 2** that AERI exhibits a more pronounced ..."

Done.

Line 504: "...European Space Agency (ESA contract 4000123417/NL/LA) for **the** HiSRAMS **development and permission for its use**, and ..."

Done.

---

## Author Comment (AC2)

In this response letter, our responses are labeled in blue. Revised sentences in the updated manuscript are shown in *italic purple text*. All line numbers correspond to the revised manuscript with tracked changes, where deletions are shown in red and additions are shown in blue.

This is a well written paper. The analysis is thorough and supports the conclusions made. The presentation quality is good. The authors first conduct AERI and HiSRAMS retrievals for ground-based observations of clear sky only, which compares the capability of the two instruments in terms of information content on temperature and water vapor. Then they demonstrate a joint retrieval between ground-based AERI and air-borne HiSRAMS clear sky and show the benefit of combining the two instruments. I don't think this paper delivers new science, but it does present a retrieval algorithm for atmospheric temperature and water vapor sounding from IR and MW. I have the following comments for authors to consider:

We thank the reviewer for the valuable comments that improve the quality of our manuscript. We have revised the manuscript accordingly.

In terms of the new science in our manuscript, our intention is to introduce the role and ability of new technology in hyperspectral microwave instruments to retrieve temperature and water vapor vertical profiles. HiSRAMS is one of the first developed hyperspectral microwave radiometer. We have included a sentence on Lines 72-73: *"With the development of HiSRAMS, we aim to demonstrate the capabilities of new technology in hyperspectral microwave instruments for retrieving temperature and water vapor vertical profiles under both clear-sky and all-sky conditions."*.

1. The value of this paper is on combining air-borne microwave and ground-based infrared sounding measurements, not much on whether or not the MW and IR combined could provide a better "all-sky" retrievals. I think it is better to point this out in your title, abstract, and literature survey, and center them on your major findings.

As one of the first hyperspectral microwave radiometers developed, the HiSRAMS instrument has provided some of the earliest data ever obtained in this study. While the strength of microwave radiometers lies in their ability to penetrate clouds, thus enhancing all-sky temperature and water vapor profiling, it is crucial first to evaluate the instrument under well-controlled conditions. Therefore, we have chosen to conduct initial tests in clear-sky environments to showcase the capabilities and significance of this new hyperspectral microwave technology. It's worth noting that ground-based, airborne, and combined hyperspectral infrared and microwave measurements each offer unique benefits due to their distinct geometric configurations.

In order to make it clearer, we have revised the sentences on Lines 12-17 in the abstract: *"In contrast to infrared radiometry, microwave radiometry can penetrate clouds, making it a valuable tool for all-sky thermodynamic retrievals. Recent advancements have led to the fabrication of a hyperspectral microwave radiometer: the High Spectral Resolution Airborne Microwave Sounder (HiSRAMS). This study utilizes HiSRAMS to retrieve atmospheric temperature and water vapor profiles under clear-sky conditions; this is an initial assessment of one of the first hyperspectral microwave radiometers, comparing the results to those from an infrared hyperspectrometer, the Atmospheric Emitted Radiance Interferometer (AERI)."*.

2. The retrieved profiles agree with the radiosonde surprisingly well, but clear differences with the first guess can be seen. Are these results all for a single case? How does the algorithm perform in general? It is hard to reach conclusions based on a single case.

Yes, the retrievals are all based on a single case. We have revised the sentence in the Conclusions and discussion section on Lines 514-517: *"Additionally, the retrieval comparison in this study relies on limited samples from a single campaign **which only provides one case study for each retrieval configuration**, thus bounding the usefulness of the error statistics and comprehensiveness of this assessment. Specifically, **only a single radiosonde** was launched during the field campaign, which may have induced temporal and spatial variability in the truth profile."*.

We have not yet developed an Observing System Simulation Experiment (OSSE) for retrievals based on "sandwich" measurements. However, OSSEs and operational retrievals for single HiSRAMS and AERI measurements have been well studied, with results similar to ours (Bliankinshtein et al., 2019; Bliankinshtein et al., 2023; Turner et al., 2014; Turner et al., 2018; Loveless, 2021). Operational retrieval algorithms for HiSRAMS and AERI under all-sky conditions are currently in development.

Bliankinshtein, N., Gabriel, P., Huang, Y., Wolde, M., Olvhammar, S., Emrich, A., Kores, M., and Midthassel, R.: Airborne Measurements of Polarized Hyperspectral Microwave Radiances to Increase the Accuracy of Temperature and Water Vapor Retrievals: an Information Content Analysis, AGU Fall Meeting, 2019.

Bliankinshtein, N., Liu, L., Gabriel, P., Xu, S., Bala, K., Wolde, M., Huang, Y., Auriacombe, O., Krus, M., and Angevain, J.-C.: Airborne validation of HiSRAMS atmospheric soundings, IGARSS IEEE International Geoscience and Remote Sensing Symposium, 4372-4375, 2023.

Loveless, D. M.: Developing a Synergy Between Space-based Infrared Sounders and the Ground-based Atmospheric Emitted Radiance Interferometer (AERI) to Improve

Thermodynamic Profiling of the Planetary Boundary Layer, The University of Wisconsin-Madison, 2021.

Turner, D. D. and Blumberg, W. G.: Improvements to the AERIoe thermodynamic profile retrieval algorithm, IEEE Journal of Selected Topics in Applied Earth Observations and Remote Sensing, 12, 1339-1354, 2018.

Turner, D. D. and Löhnert, U.: Information content and uncertainties in thermodynamic profiles and liquid cloud properties retrieved from the ground-based Atmospheric Emitted Radiance Interferometer (AERI), Journal of Applied Meteorology and Climatology, 53, 752-771, 2014.

3. The retrieval bias around the sharp vertical features is very large, which is due to the lack of vertical resolution. This could be demonstrated by calculating the bias by comparing retrievals with the radiosondes that are vertically smoothed by the averaging kernel. Perhaps it is more interesting to ask what kind of instrument designs could resolve such feature, new channels, lower measurement error, etc.

We applied Eq. R2-1 to smooth the retrieved temperature and water vapor profiles, and the updated retrieval bias figures are shown in Figures R2-1 and R2-2. To some extent, the sharp vertical features have been reduced. However, relatively sharp biases, primarily caused by fine vertical features, are still present. To accurately display the biases and avoid confusion, we have decided to retain the original Figures 8 and 13. The error analysis using the smoothed profiles has been included in the Supplementary Information.

We agree that understanding what kinds of instrument designs could resolve such features is important. Based on the OSSE for HiSRAMS-only measurements (Bliankinshtein et al., 2019), the microwave absorption characteristics already limit the information we can obtain from the atmosphere. In other words, regardless of how high the spectral resolution, how wide the spectral range, or how low the measurement error within a reasonable setup, the information content from a hyperspectral microwave radiometer cannot be significantly increased. A more detailed analysis of combining microwave and infrared hyperspectral radiometers warrants further investigation in the future.

$$x_{truth}^{smoothed} = A(x_{truth} - x_a) + x_a \qquad \text{Eq. R2-1}$$

[Figure]

Figure R2-1: Comparison of retrieval bias and uncertainty in ground-based retrievals for (a) temperature and (b) water vapor. The truths used to determine the bias are smoothed using Eq. R2-1.

[Figure]

Figure R2-2: Comparison of retrieval bias and uncertainty in joint retrievals for (a) temperature and (b) water vapor. The truths used to determine the bias are smoothed using Eq. R2-1.

Bliankinshtein, N., Gabriel, P., Huang, Y., Wolde, M., Olvhammar, S., Emrich, A., Kores, M., and Midthassel, R.: Airborne Measurements of Polarized Hyperspectral Microwave Radiances to Increase the Accuracy of Temperature and Water Vapor Retrievals: an Information Content Analysis, AGU Fall Meeting, 2019.

---

## Author Comment (AC3)

In this response letter, our responses are labeled in blue. Revised sentences in the updated manuscript are shown in *italic purple text*. All line numbers correspond to the revised manuscript with tracked changes, where deletions are shown in red and additions are shown in blue.

This paper is well written, with good explanations of the OEM methodology and analysis that supports the conclusions. One could argue that the Optimal Estimation methodology is well known, but the paper has practical examples that accompany the explanations.

Having said that, not much new or unexpected conclusions are in this paper, but it can find a place in an Atmospheric Measurements and Techniques paper if some fixes are made.

We thank the reviewer for the valuable comments that improve the quality of our manuscript. We have revised the manuscript accordingly.

In terms of the new or unexpected conclusions in our manuscript, our intention is to introduce the role and ability of new technology in hyperspectral microwave instruments to retrieve temperature and water vapor vertical profiles. We have included a sentence on Lines 72-73: *"With the development of HiSRAMS, we aim to demonstrate the capabilities of new technology in hyperspectral microwave instruments for retrieving temperature and water vapor vertical profiles under both clear-sky and all-sky conditions."*.

1) Line 470 : it is very hard to tell if the results presented truly are statistical N >> 1, or if only one case is done for the instruments alone on the ground, and one other case is done for the HISRAMS/AERI synergy. Your conclusions are too general if only one case is considered, especially since what you find is not surprising.

(AERI and HISRAMS together give "better" results (closer to truth) than one instrument alone, and a top/bottom sandwich brings a good deal of information together).

In our case study, N equals 1. The conclusion remains valid when determining the Degree of Freedom of the Signal (DFS) and the posterior uncertainty, as $\mathbf{K}$, $\mathbf{S_a}$, and $\mathbf{S_e}$ usually do not change significantly between cases. However, the comparison of the retrieved results with the truth is based solely on the presented case study.

We have revised the sentence in the Conclusions and discussion section on Lines 514-517: *"Additionally, the retrieval comparison in this study relies on limited samples from a single campaign **which only provides one case study for each retrieval configuration**, thus bounding the usefulness of the error statistics and comprehensiveness of this assessment. Specifically, **only a single radiosonde** was launched during the field campaign, which may have induced temporal and spatial variability in the truth profile."*.

2) Similarly, are you really doing an clear-sky retrieval. How do you know there are no clouds? Your paper gets confusing at the beginning when you mention "clear sky" in the title and then talk about the use of HISRAMS in allsky conditions. Can you clarify?

Thanks to the new technology of hyperspectral microwave radiometers, their primary advantage is the ability to penetrate clouds, which helps improve all-sky temperature and water vapor profiling. However, as this is a new technology, it is important to first test the instrument under well-controlled conditions, for which we have selected clear-sky environments. Thus, in the Introduction section, we have revised the sentence on Lines 73-75: *"While the primary advantage of microwave radiometers lies in their ability to retrieve in cloudy-sky conditions, this study focuses initially on clear-sky retrievals **under well-controlled conditions to ensure the instrument's performance."***.

In terms of determining whether the sky condition is clear, we have presented the radiosonde profile along with the clear-sky radiative closure analysis results in Liu et al. (2024). The personnel who collected the data also confirmed that no visible clouds were present that day. Absence of clouds below 6.8 km was also confirmed with airborne in-situ probes. The possibility of sub-visible cirrus is discussed in Liu et al. (2024), and it is considered extremely low.

Liu, L., Bliankinshtein, N., Huang, Y., Gyakum, J. R., Gabriel, P. M., Xu, S., and Wolde, M.: Radiative closure tests of collocated hyperspectral microwave and infrared radiometers, Atmospheric Measurement Techniques, 17, 2219-2233, 2024.

3) Figure 2,6 : Please show mean(observations - calculations) and std(observations - calculations) for both instruments. You could be getting great results while having poor spectral fits (ie the bias and std. dev are larger than the NeDT of the instruments).

In particular could you indicate the surface channels of the HISRAMS when showing the biases?

Since we have only a single case study for different geometric configurations, we presented the observed versus simulated spectra for various case studies in Figures S3-1 and S3-2. In most channels, the differences between observations and final retrieved simulations fall within the $3\sigma$ uncertainty range. However, exceptions include some spike channels in the ground-based AERI retrieval and the weak absorption channels (around 52 GHz) in the ground-based HiSRAMS retrieval. Liu et al. (2024) analyzed the radiative closure for both HiSRAMS and AERI, noting a systematic bias around 52 GHz in HiSRAMS. In ground-based HiSRAMS zenith-pointing measurements, the surface channels correspond to strong absorption channels located between 54 and 58 GHz within the oxygen band.

We have included this analysis in the Supplement Information.

[Figure]

Figure S3-1: Radiance or brightness temperature differences between observations and the final retrieved simulation for ground-based retrievals: (a) AERI; (b) HiSRAMS oxygen band; (c) HiSRAMS water vapor band. The 1σ uncertainties are determined from the square root of the diagonal components of $S_e$.

[Figure]

Figure S3-2: Radiance or brightness temperature differences between observations and the final retrieved simulation for joint retrievals: (a) AERI; (b) HiSRAMS oxygen band; (c) HiSRAMS water vapor band. The 1σ uncertainties are determined from the square root of the diagonal components of $S_e$.

Liu, L., Bliankinshtein, N., Huang, Y., Gyakum, J. R., Gabriel, P. M., Xu, S., and Wolde, M.:
Radiative closure tests of collocated hyperspectral microwave and infrared radiometers,
Atmospheric Measurement Techniques, 17, 2219-2233, 2024.

4) The paper could be much better organized. For example instead of "sandwich" you could use the more traditional "synergy" term.

The term "synergy" refers to the combination of different instruments. "Sandwich," on the other hand, implies the combination of both zenith-pointing and nadir-pointing measurements from different instruments simultaneously. Thus, we have decided to retain the use of the term "sandwich" measurements.

5) Figures S5 - S8 are ..?

Figures S1 to S7 are provided in the supplemental document, which can be accessed at https://doi.org/10.5194/egusphere-2024-1045-supplement.

6) Lines 155-162, and lines 174-180, should be combined together rather than being separated. Then again line 189 the n_{level} is mentioned.

We have combines these two paragraphs together with the description of the matrices for joint retrievals on Lines 173-189: "In this study, we retrieve temperature and water vapor vertical profiles simultaneously using single instruments (AERI or HiSRAMS) and joint instruments (AERI and HiSRAMS) respectively. Thus, $x$ equals to $\begin{bmatrix} x_T \\ x_q \end{bmatrix}$ with a dimension of $38 \times 2 = 76$.

For all retrieval cases, the dimensions of the matrices $S_a$, $S$, and $A$ are based solely on the dimension of the vertical level ($n_{level} \times n_{level}$), maintaining a consistent structure with the upper-left sub-matrix for temperature and the lower-right sub-matrix for water vapor. This structure allows us to separate the information of temperature and water vapor. Because HiSRAMS is an airborne instrument, its observational capabilities can be limited by altitude, affecting $n_{level}$ for different case studies when it is nadir-pointing. In order to test the full potential of AERI and HiSRAMS to retrieve temperature and water vapor concentration profiles, all the instrumental channels are kept, with the result that $n_{AERI} = 2490$ and $n_{HiSRAMS} = 2850$ (including the measurements of both spectrometers of HiSRAMS). When retrieving the temperature and water vapor vertical profiles using either AERI or HiSRAMS alone, the dimensions of $S_e$ and $K$ are $n_{instrument} \times n_{instrument}$ and $n_{instrument} \times n_{level}$, respectively, where 'instrument' refers to either AERI or HiSRAMS. For joint retrieval:

$$y_{joint} = \begin{bmatrix} y_{AERI} \\ y_{HiSRAMS} \end{bmatrix} \tag{8}$$

$$S_{e,joint} = \begin{bmatrix} S_{e,AERI} & 0 \\ 0 & S_{e,HiSRAMS} \end{bmatrix} \tag{9}$$

$$K_{joint} = \begin{bmatrix} K_{AERI} \\ K_{HiSRAMS} \end{bmatrix} \qquad (10)$$

The dimensions of $y_{joint}$, $S_{e,joint}$, and $K_{joint}$ are $(n_{AERI} + n_{HiSRAMS}) \times 1$, $(n_{AERI} + n_{HiSRAMS}) \times (n_{AERI} + n_{HiSRAMS})$, and $(n_{AERI} + n_{HiSRAMS}) \times n_{level}$ respectively.".

7) Similarly lines 434 to 439 could be moved to the conclusion

This paragraph summarizes Section 5: Joint airborne HiSRAMS and ground-based AERI retrievals. We intend to provide summary paragraphs for both Section 4 (Lines 356-360) and Section 5 (Lines 478-483). In the Conclusion and Discussion section, we have also briefly summarized our work, findings, and the limitations of the analysis.

---

## Author Response (AR2)

In this response letter, our responses are labeled in blue. Revised sentences in the updated manuscript are shown in *italic purple text*. All line numbers correspond to the revised manuscript with tracked changes, where deletions are shown in red and additions are shown in blue.

2 very minor improvements are suggested before publication:

We thank Dr. Lin for the comments, which have further enhanced the clarity of our manuscript. We have revised the manuscript accordingly.

(1) Table 1: Please add a reference where more information concerning the Radiosonde data is described (Liu et al. (2024)).

In Table 1 (Line 90), we have added an additional sentence to the description of the radiosonde data: "*For more detailed information about the radiosonde sounding, refer to Liu et al. (2024).*"

(2) Line 126, "Figure S1": For better clarity, please add "(see Supplementary Information)".

We have added this clarification in Line 121.